# Flexible Magnifying Endoscopy with Narrow Band Imaging for Diagnosing Uterine Cervical Neoplasms: A Multicenter Prospective Study

**DOI:** 10.3390/jcm10204753

**Published:** 2021-10-16

**Authors:** Hideki Kobara, Kunihisa Uchita, Noriya Uedo, Jun Kunikata, Kenji Yorita, Naoya Tada, Noriko Nishiyama, Yuriko Shigehisa, Chihiro Kuroiwa, Noriko Matsuura, Yohei Takahashi, Yuka Kai, Uiko Hanaoka, Yumiko Kiyohara, Shoji Kamiura, Kenji Kanenishi, Tsutomu Masaki, Koki Hirano

**Affiliations:** 1Department of Gastroenterology and Neurology, Faculty of Medicine, 1750-1 Ikenobe, Miki, Kita, Kagawa 761-0793, Japan; n-tada@med.kagawa-u.ac.jp (N.T.); n-nori@med.kagawa-u.ac.jp (N.N.); tmasaki@med.kagawa-u.ac.jp (T.M.); 2Department of Gastroenterology, Kochi Red Cross Hospital, 2-13-51 Shinhonmachi, Kochi 780-8562, Japan; ucchy31@yahoo.co.jp (K.U.); yurikonoguchi630@gmail.com (Y.S.); kuroiwa.kochi@gmail.com (C.K.); 3Department of Gastrointestinal Oncology, Osaka International Cancer Institute, 3-1-69 Otemae, Chuo-ku, Osaka 541-8567, Japan; noriya.uedo@gmail.com (N.U.); ibura9@yahoo.co.jp (N.M.); 4Department of Clinical Research Support Center, Faculty of Medicine, Kagawa University, 1750-1 Ikenobe, Miki, Kita, Kagawa 761-0793, Japan; kunikata@med.kagawa-u.ac.jp; 5Department of Diagnostic Pathology, Kochi Red Cross Hospital, 2-13-51 Shinhonmachi, Kochi 780-8562, Japan; kenjiyorita@gmail.com; 6Department of Gynecology, Kochi Red Cross Hospital, Kochi, 2-13-51 Shinhonmachi, Kochi 780-8562, Japan; yoheitaka@gmail.com (Y.T.); memento-mori@mbr.nifty.com (Y.K.); hirarin0908@k7.dion.ne.jp (K.H.); 7Department of Perinatology and Gynecology, Faculty of Medicine, Kagawa University, 1750-1 Ikenobe, Miki, Kita, Kagawa 761-0793, Japan; uhanaoka@med.kagawa-u.ac.jp (U.H.); kane@med.kagawa-u.ac.jp (K.K.); 8Department of Gynecology, Osaka International Cancer Institute, 3-1-69, Otemae, Chuo-ku, Osaka 541-8567, Japan; kiyoyumi000@hotmail.co.jp (Y.K.); kamiura-sh@mc.pref.osaka.jp (S.K.); 9Department of Obstetrics and Gynecology, Japan Community Health Care Organization Osaka Hospital, 4-2-78 Fukushima, Fukushima-ku, Osaka 553-0003, Japan

**Keywords:** cervical intraepithelial neoplasia, colposcopy, endoscopy, uterine cervical neoplasms

## Abstract

We aimed to investigate the diagnostic ability of magnifying endoscopy with narrow band imaging (ME-NBI) for cervical intraepithelial neoplasia grade 2 or worse (CIN2+). This was a multicenter prospective study. Eligible patients had positive Pap smear results or follow-up high-grade cytology or CIN3 diagnosed in referring hospitals. Patients underwent ME-NBI by a gastrointestinal endoscopist, followed by colposcopy by a gynecologist. One lesion with the worst finding was considered the main lesion. Punch biopsies were collected from all indicated areas and one normal area. The reference standard was the highest histological grade among all biopsy specimens. The primary endpoint was the detection rate of patients with CIN2+ in the main lesion. The secondary endpoints were diagnostic ability for CIN2+ lesions and patients’ acceptability. We enrolled 88 patients. The detection rate of ME-NBI for patients with CIN2+ was 79% (95% CI: 66–88%; *p* = 1.000), which was comparable to that of colposcopy (79%; *p* = 1.000). For diagnosing CIN2+ lesions, ME-NBI showed a better sensitivity than colposcopy (87% vs. 74%, respectively; *p* = 0.302) but a lower specificity (50% vs. 68%, respectively; *p* = 0.210). Patients graded ME-NBI as having significantly less discomfort and involving less embarrassment than colposcopy. ME-NBI did not show a higher detection ability than colposcopy for patients with CIN2+, whereas it did show a better patient acceptability.

## 1. Introduction

Uterine cervical cancer is caused by human papillomavirus (HPV) infection, and proper screening programs promise cancer detection at an early stage. Patients with cervical intraepithelial neoplasia grade 3 (CIN3) or intraepithelial cancer can be treated by cervical conization, which preserves fertility by avoiding hysterectomy.

The World Health Organization is establishing a strategy to achieve a one-third reduction in premature mortality from cervical cancer by 2030 [1]. The strategy emphasizes the importance of triple intervention targets constituting increased HPV vaccination rates, screening rates, and treatment of preinvasive and invasive lesions for cervical cancer. Recently, in many screening programs, HPV testing has been adopted as the first triage, followed by colposcopy [2]. Meanwhile, traditional screening constitutes cytological examination (Pap smear) followed by colposcopy. In both methods, colposcopy is important for identifying lesions with CIN grade 2 or worse (CIN2+) and for treatment decision making. A recent quality-controlled review found a moderate sensitivity of colposcopy for CIN2+ of 75.1%, a specificity of 71.0%, and a positive predictive value of 72.0% [3], suggesting that the diagnostic accuracy of colposcopy-directed punch biopsies needs to be improved. The reasons for the low diagnostic accuracy of conventional colposcopy are the low image quality and a limited visual field when this method is used with a Cusco speculum. Moreover, although colposcopic examination is tolerable for most women, some patients feel pain, cold, or discomfort during insertion of the speculum [4]. Therefore, developing accurate and comfortable new methods for diagnosing cervical cancer is desired.

Currently, an image-enhanced endoscopy technology, narrow band imaging (NBI), plays an important role in the diagnosis of gastrointestinal neoplasms [5]. NBI uses narrow band 415 nm and 540 nm wavelength illumination lights, which are well reflected on the mucosal surface and are strongly absorbed by hemoglobin, contrasting surface structures and the mucosal vasculature [6]. Combining magnifying endoscopy with narrow band imaging (ME-NBI) allows clear visualization of the micro-structure and microvessels of the superficial mucosa that correspond to histology. In particular, the effectiveness of ME-NBI for detecting and characterizing esophageal squamous cell carcinoma has been demonstrated in several studies [7,8,9,10]. Therefore, we expected that ME-NBI would also be useful for diagnosing cervical CINs. Although our previous investigation suggested the potential efficacy of this method for diagnosing cervical CINs [11,12], there is minimal evidence to support the usefulness of ME-NBI. The aim of this study was to examine the diagnostic ability of ME-NBI for diagnosing CIN compared with that of colposcopy.

## 2. Patients and Methods

### 2.1. Study Design

This was a multicenter prospective study conducted in Kagawa University Hospital, Kochi Red Cross Hospital, and Osaka International Cancer Institute. Our primary aim was to prospectively evaluate the detection ability of ME-NBI for patients with CIN2+. However, we considered that it was necessary to offer colposcopic examination as a standard diagnostic examination to the study participants. Moreover, we wanted to have a comparator for the diagnostic ability of ME-NBI. Therefore, this study was designed as a non-randomized, paired comparison study between ME-NBI and colposcopy.

Initially, patients underwent ME-NBI examination by a gastrointestinal endoscopist in an endoscopy unit. Then, the patients were moved to a gynecology unit, and received colposcopic examination by a gynecologist who was blinded to the ME-NBI findings. At least a 30-min interval was maintained between the two examinations to extinguish the aceto-whitening effect on the mucosa that occurs during ME-NBI. In each examination, one lesion with the worst finding (the most obvious white epithelium or the most irregular vessels) was identified as the main lesion. Immediately after each examination, the location and findings for all detected lesions were documented in a dedicated case report that included an illustration sheet of the uterine cervix. Finally, the gynecologist performed punch biopsies for all indicated lesions on both the ME-NBI and colposcopy illustration sheets. One random biopsy specimen was taken from a normal area as a negative control. In cases of no abnormal findings in both examinations, one random biopsy was performed in any location. The reference standard was defined as the highest histological grade among all biopsy specimens.

The study protocol was approved by the Clinical Ethics Committee in Kagawa University on 26 April 2018 (No. H30-061, Kagawa, Japan), and in each facility in accordance with the Declaration of Helsinki. The use of a gastroscope for diagnosis of the uterine cervix neoplasm was approved by the institutional review board. The study was registered in the University Hospital Medical Information Network Clinical Trials Registry as UMIN 000033189. The manuscript was written according to the standards for reporting diagnostic accuracy (STARD) initiative [13].

### 2.2. Participants

#### Inclusion and Exclusion Criteria

Consecutive patients with one of the following conditions were assessed for eligibility: a positive Pap smear test result, follow-up for high-grade cytology, and diagnosis of CIN3 in referring hospitals. The inclusion criteria were age between 20 and 64 years and provision of written informed consent. Exclusion criteria were patients with a history of uterine cervical operation, mental illness or symptoms, or who were pregnant or possibly pregnant, and patients judged inappropriate for study participation by the attending physician.

### 2.3. Procedures and Evaluations

#### 2.3.1. ME-NBI Procedure

The NBI system (EVIS LUCERA ELITE; Olympus Medical Systems, Co., Ltd., Tokyo, Japan) used in this study consisted of a magnifying gastrointestinal video endoscope (GIF-H260Z or -H290Z), an image processor (CV-290), and a light source (CLV-290). A soft black hood (MAJ1990; Olympus Medical Systems, Co., Ltd.) and an occlusion balloon (Fuji Systems Corporation, Tokyo, Japan) were mounted to the videoendoscope (Figure 1).

A carbon dioxide insufflation device (UCR, Olympus Medical Systems, Co., Ltd.) was used for all endoscopic procedures. The patients underwent ME-NBI examination in the left lateral decubitus position (Figure 2).

The videoendoscope was inserted into the vagina, and the vaginal orifice was occluded by the balloon to achieve sufficient air insufflation and water immersion. Adherent mucus was removed using the scope’s water jet. The squamocolumnar junction was circumferentially observed at long and middle distances in white light images, and subsequently in NBI images. Suspicious areas were then inspected at a close distance with ME-NBI (maximum magnification: ×85). Finally, 3% acetic acid (20–30 mL) was applied through the scope’s working channel using a syringe. The uterine cervix was immersed in acetic acid for 1 min, and was then observed under white light and NBI.

The ME-NBI findings were evaluated by the presence of thin- or thick-white epithelium and irregular intraepithelial papillary capillary loops (IPCL) [11,12]. White epithelium was defined as well-demarcated epithelium that was whiter than the surrounding epithelium, and was classified as thin- and thick-white epithelium according to the visibility of the underlying vessel. Atypical IPCL was defined as a microvessel that showed more than two of the following four findings: dilatation, meandering, caliber change, and non-uniformity according to the IPCL classification in the esophagus [10]. The acetowhite findings observed in acetic acid endoscopy conformed to the following terminology used for colposcopy [14]: thin acetowhite epithelium (W1) and dense acetowhite epithelium (W2). The diagnostic criteria for CIN2+ in ME-NBI were either the presence of thick-white epithelium or thin-white epithelium plus atypical IPCLs in NBI (Figure 3), or dense acetowhite epithelium (W2) in acetic acid endoscopy [12].

All procedures were performed by a single examiner among four endoscopists with experience performing ME-NBI examinations for more than 600 cases of gastrointestinal neoplasms (HK, KU, NU and NN). Interpretation of ME-NBI findings of the uterine cervix was trained using an image atlas before study participation.

#### 2.3.2. Colposcopic Procedure

Patients underwent colposcopic examination in the lithotomy position in a gynecology unit (Figure 2). After visualizing the uterine cervix using a Cusco speculum (SANRITU, Tokyo, Japan), a colposcopic instrument (ZEISS Colposcope KSK 150 FC, Carl Zeiss Meditec AG, Jena, Germany), which allows a maximum magnifying observation of ×21.5, was used for examination. Colposcopic examination was performed after 3% acetic acid application.

Colposcopic findings were evaluated according to terminology described in the Rio 2011 Colposcopy Nomenclature of the International Federation for Cervical Pathology and Colposcopy [14]. According to the most frequent abnormal colposcopic findings, i.e., acetowhite epithelium, mosaic, punctation, and other abnormal findings, lesions were categorized as follows: Grade 1 (minor), thin acetowhite epithelium, fine punctation, fine mosaic; Grade 2 (major), dense acetowhite epithelium, coarse punctation, coarse mosaic; Suspicious for invasion, atypical vessels, fragile vessels, irregular surface, exophytic lesion, necrosis ulceration (necrotic), tumor, or gross neoplasm. Lesions of Grade 2 or worse were diagnosed as CIN2+ in colposcopy.

Each colposcopic procedure was performed by a single experienced gynecologist (YK, UH, KK or KH).

### 2.4. Patients’ Acceptability and Adverse Events

After both examinations, patients were asked to grade discomfort and embarrassment for each examination from 1 to 5, and to indicate their preference out of the two methods for the next examination. Punch biopsies were performed only in the colposcopy group. To maintain equality between the methods, the questionnaire excluded pain accompanied by the punch biopsy, but did include discomfort associated with colposcopy. This information was provided on an unsigned self-completed questionnaire sheet without the attendance of medical staff.

Adverse events were evaluated at the end of the protocol examination and at the time of consultation according to the Common Terminology Criteria for Adverse Events (CTCAE) ver. 5.0 [15].

### 2.5. Histological Examination

Punch biopsy samples were fixed in 10% neutral-buffered formalin, processed into paraffin-embedded blocks, sectioned for 3-μm slices, and stained with hematoxylin and eosin. The histological diagnosis was made by experienced pathologists in each institution according to the General Rules for Clinical and Pathological Management of Uterine Cervical Cancer, edited by the Japan Society of Obstetrics and Gynecology and the Japanese Society of Pathology [16]. The pathologists were blinded to the ME-NBI and colposcopy findings.

### 2.6. Outcome Measures

The primary endpoint was the detection ability of ME-NBI for patients with CIN2+ in a main lesion biopsy. This was calculated as the proportion of histological CIN2+ in the main lesion biopsy to histological CIN2+ in the reference standard (highest histological grade among all biopsy specimens) (Figure 4).

A key secondary endpoint was the diagnostic ability of ME-NBI and colposcopy for histological CIN2+ lesions. Sensitivity was calculated as the proportion of correct diagnoses (CIN2+ diagnosis in the index test) among histological CIN2+ in the main lesion biopsy (Figure 4). Specificity was calculated as the proportion of correct diagnoses (non-CIN2+ diagnosis in the index test) among histological non-CIN2+ in the main lesion biopsy. Other secondary endpoints were the proportion of cases in which the entire circumferential transition zones of the squamocolumnar junction were visible, the proportion of cases in which the external uterine orifice was visible, the incidence of adverse events, and patients’ acceptability.

### 2.7. Statistical Methods

The case report forms for ME-NBI and colposcopy were faxed to the Department of Clinical Research Support Center, Faculty of Medicine, Kagawa University on the day of the examination. The histological examination findings for the biopsy specimens were faxed within 7 days after the pathological report. All analyses were performed by an independent clinician (JK) in the Clinical Research Support Center (Kagawa, Japan).

Categorical variables were presented as numbers and percentages, and continuous variables as medians and ranges. Diagnostic values were presented as proportions and 95% confidence intervals (CI). Categorical variables were compared using McNemar’s test, but the key secondary endpoint was analyzed using the Chi square test because the main lesions detected by ME-NBI and colposcopy were not the same. The patients’ acceptability grades were compared using the asymptotic Wilcoxon signed rank test. All statistical analyses were performed using R version 3.6.1 (R Foundation for Statistical Computing, Vienna, Austria; 2019), and two-tailed *p* < 0.05 was considered significant.

The sample size was calculated according to the width of the 95% CI of the primary endpoint. The detection rate of patients with CIN2+ was assumed to be 70% [3,17]. In our group discussion, we considered that a ±13% range of 95% CI for the detection rate would be acceptable; therefore, 48 patients with CIN2+ were required. The proportion of CIN2+ in patients with positive Pap smear test results or follow-up for high-grade cytology was estimated as 40%, and that in patients with CIN3 was estimated as 90%. Finally, we calculated the required number of patients to be 95 when the number of patients with CIN3 was 20 according to our recent clinical records (among the required 48 patients with CIN2+, 18 were from 20 patients with a diagnosis of CIN3 in referring hospitals, and 30 were from 75 patients with positive Pap smear test results or follow-up for high-grade cytology).

## 3. Results

### 3.1. Patient Enrollment

A total of 95 patients were enrolled between September 2018 and June 2020 in the three institutions. A total of 7 of the 95 patients were excluded because the findings for their index tests for the main lesions were not described on the case reporting form; the remaining 88 patients were included for the final analysis (Figure 5).

The patients’ demographics are summarized in Table 1. The patients were women with a median age of 40 years (range: 21–67 years) who received examinations for positive Pap smear test results (*n* = 72), follow-up for high-grade cytology (*n* = 13), and a CIN3 diagnosis in a referring hospital (*n* = 3). Twenty-five patients had HPV infection, and five had a history of HPV vaccination.

### 3.2. Outcomes

The final histological diagnoses in the reference standard comprised non-cancerous lesions (*n* = 8), CIN1 (*n* = 32), CIN2 (*n* = 13), CIN3 (*n* = 34), and microinvasive carcinoma (*n* = 1).

The detection rate of ME-NBI for patients with CIN2+ was 79.2% (95% CI: 65.7–88.3%), which was not significantly higher than that of colposcopy (79.2%, *p* = 1.000) (Table 2).

Regarding the diagnostic ability of ME-NBI and colposcopy for histological CIN2+ lesions (Figure 6), ME-NBI tended toward a high sensitivity compared with colposcopy (86.8% vs. 73.7%, respectively; *p* = 0.150), whereas its specificity was lower (50.0% vs. 68.0%, respectively; *p* = 0.067). Thus, no obvious difference between the two methods was observed in the overall accuracy (65.9% vs. 70.5%, ME-NBI vs. colposcopy, respectively; *p* = 0.517). The proportion of patients with visible entire circumferential transition zones was significantly higher with ME-NBI than with colposcopy (97.7% vs. 90.9%, respectively; *p* = 0.031).

No serious (≥Grade 2 in CTCAE ver. 5.0) adverse events were observed. Among patients who answered the questionnaire, 96.6% graded their discomfort during the procedure as 1 or 2 for ME-NBI, while 57.5% graded their discomfort during colposcopy as 4 or 5. Regarding embarrassment, the proportion of patients reporting grade 4 or 5 was 7.7% for ME-NBI, compared with 46.0% for colposcopy. As a next examination, 65 patients preferred ME-NBI (75.6%), 2 preferred colposcopy (2.3%), and 19 reported that they would accept both methods (22.1%) (ME-NBI vs. colposcopy, *p* < 0.001).

## 4. Discussion

In this multicenter prospective study, we found two important aspects of ME-NBI for diagnosing CIN. First, ME-NBI did not significantly show a higher detection ability than colposcopy for patients with CIN2+. Second, patients’ acceptability of ME-NBI was significantly better than for colposcopy.

Although whether cytology or HPV testing is an appropriate tool for initial screening for cervical cancer is controversial [2,18], among patients with abnormal findings in screening tests, colposcopy plays a pivotal role in detecting and characterizing premalignant lesions or early-stage cervical cancer. However, two previous meta-analyses indicated that colposcopy had a wide variation in sensitivity (30–99% and 65.9–100%, respectively) and specificity (39–93% and 5–80%, respectively) for diagnosing CINs [19,20]. Most studies appeared to be subject to bias, especially verification bias, which is problematic for studies evaluating the diagnostic accuracy of colposcopy. Moreover, as these studies calculated sensitivity and specificity according to different reference standards, such as punch biopsies or excisional biopsies following punch biopsies, it is difficult to compare the diagnostic accuracy of ME-NBI with the results in these studies. Therefore, in this study, we prepared a paired control arm of colposcopy using the same reference standard. Moreover, we compared the diagnostic ability of two methods for the single main lesion with the worst appearance. In addition, one random biopsy was also performed in a normal area to serve as a negative control. These efforts to standardize the evaluation methods confirmed the diagnostic values of ME-NBI and colposcopy obtained in this study.

In a randomized, controlled trial investigating the detection rates for patients with superficial squamous cell carcinoma in the head and neck region and the esophagus, ME-NBI was significantly superior to white light endoscopy (97% vs. 55%, respectively) [8]. Accordingly, we expected that the detection ability of ME-NBI for patients with CIN2+ would exceed that of colposcopy, but, in fact, it was similar. When we reviewed endoscopic and colposcopic images of eight discordant cases between ME-NBI and colposcopy, four ME-NBI true positive and colposcopy false negative cases included difficulty in observing the entire circumferential transition zone using a Cusco speculum (*n* = 1), regression of the transition zones toward the cervical orifice (*n* = 1), no acetowhite findings (*n* = 1), and a lesion hidden behind a benign polyp (*n* = 1). In these cases, ME-NBI could achieve better views of the entire uterine cervix vs. colposcopy, and could make a correct diagnosis. In contrast, three of the four colposcopy true positive and ME-NBI false negative cases were caused by sampling error in the colposcopic biopsy. Although the locations of all lesions detected in ME-NBI were indicated on the illustration sheet, in some cases, the punch biopsy sample was not taken from the exact area of a main lesion, which reduced the true positive rate. This was partly associated with different maneuvers for observation, i.e., the expansion of the vagina with air or water in ME-NBI compared with an instrumental opening with Cusco’s speculum in colposcopy. Moreover, in some cases, the location of the main lesion in ME-NBI was mistaken by gynecologists because the orientation axis was different between the left lateral decubitus position in ME-NBI and the lithotomy position in colposcopy. In contrast, in colposcopy, a punch biopsy specimen was taken from the exact area immediately after observation.

In the per-lesion analysis, we investigated the overall diagnostic performance of ME-NBI and colposcopy for CIN2+ lesions. Compared with colposcopy, ME-NBI had a high sensitivity (86.8%), while its specificity was low (50.0%), resulting in a similar low overall accuracy of 65.9% and suggesting that ME-NBI was likely to overestimate a diagnosis of CIN.

In this study, we used the IPCL classification in the esophagus for the diagnostic criteria of CIN2+; however, in the uterine cervix, the presence of chronic inflammation interfered with the diagnosis of CIN according to the microvessel findings. Thus, we now assume that the microvessel findings are better used as an adjunctive finding to the acetowhite appearance for diagnosing CIN.

In the field of gastrointestinal endoscopy, artificial intelligence is now being used to diagnose superficial esophageal squamous cell carcinoma [21]. This new computer technology may refine the diagnostic algorithm of ME-NBI by using microvessel appearance and may enhance the image quality of ME-NBI. In this study, acetowhite epithelium in colposcopy had a high specificity, but its sensitivity was lower than that of ME-NBI for lesion detection. We believe that, as a tool for screening examination, the high sensitivity of ME-NBI is clinically beneficial.

Supporting the good visibility of the uterine cervix in ME-NBI, the proportion of patients with visible entire circumferential transition zones was higher with ME-NBI than with colposcopy. The visibility of the uterine cervix influences the diagnostic accuracy of CIN. The visual field obtained in colposcopy using Cusco’s speculum is sometimes limited, especially in patients with a thick cervix or in middle-aged patients with regressed transition zones. Conversely, ME-NBI, with our developed vagina-occluding balloon, enabled examiners in this study to secure a visual field that included the entire uterine cervix and vaginal fornix by air insufflation via an endoscope, without special skills.

For colposcopic examination, 57.5% of patients graded the discomfort score as “4 or 5”; however, only 1.1% of the patients indicated a score of “4 or 5” for ME-NBI. Because we considered that vaginal expansion with Cusco’s speculum might cause patients discomfort, we asked them to grade discomfort during this examination. However, patients might also have felt pain during the punch biopsy, which was performed only in the colposcopy group. Thus, to reduce inequality between the methods, the questionnaire excluded pain accompanied by the punch biopsy, but did include discomfort associated with colposcopy. Moreover, the significantly low feelings of embarrassment during ME-NBI, and the strong preference for ME-NBI for the next examination reported indicate an important need to develop new and comfortable methods of cervical cancer screening.

We consider this an exploratory study for examining the feasibility of ME-NBI for diagnosing patients with CIN. Moreover, we anticipated that actual values of the diagnostic ability of ME-NBI and colposcopy would help design future comparative studies. Although the detection ability of ME-NBI for patients with CIN2+ did not exceed that of colposcopy, the same detection rate obtained for the two methods justifies the feasibility of ME-NBI in clinical practice. The extremely high acceptability rate of ME-NBI also supports a non-inferiority study design for future trials. Furthermore, we have recently investigated the quality of tissue sampling using endoscopic biopsy forceps under ME-NBI observation, and the results showed similar diagnostic performance to the colposcopic punch biopsy for the histological diagnosis of CIN [22]. Because some of the false positive findings with ME-NBI were caused by sampling error with the punch biopsy, we are still considering the possibility of performing a superiority study of ME-NBI with endoscopic biopsy forceps over colposcopy with the punch biopsy.

Several issues are of concern regarding the generalization of ME-NBI for cervical cancer diagnosis. Basically, gynecologists are unfamiliar with the diagnosis of microvessels in ME-NBI. However, recently, the same situation has been experienced by gastrointestinal endoscopists in the diagnosis of superficial neoplasms of the gastrointestinal tract by ME-NBI. E-learning programs would improve interpretation of the findings [23], and artificial intelligence will likely steepen the learning curve of ME-NBI diagnosis for gynecologists in the future. Instruments and maneuvers for the ME-NBI diagnostic procedures are completely different between flexible gastrointestinal endoscopy and colposcopy. A short rigid endoscope with ME-NBI function may be well accepted by gynecologists because they are familiar with maneuvering rigid laparoscopes. Although ME-NBI requires a dedicated balloon device, all other required equipment is available in endoscopy units worldwide. Therefore, we assume that ME-NBI has a high possibility of being integrated into clinical practice for cervical cancer diagnosis in the future.

Further limitations in this study must be addressed. First, this was not a parallel randomized controlled trial. Although we achieved acceptable data using endoscopic biopsy forceps for tissue sampling [22], the current standard biopsy method is a punch biopsy under colposcopy. Therefore, we determined that, in a randomized controlled design, patients assigned to the ME-NBI arm had to undergo colposcopy as well, making the study protocol complex. Because both examinations were performed by totally different evaluators in different rooms, we consider that this sequential paired comparison design provided sufficient information for an exploratory study. Second, the patients’ acceptability of the two diagnostic methods was evaluated in Japanese women. Acceptance of medical examination, balanced with its discomfort, is affected by many factors, such as social education, tradition, and culture. These issues must be considered when interpreting our results. Third, the colposcopic green-filter for evaluating neoplastic vascularization was not introduced in this study because it was not clinically used in our institutions. A comparative study between ME-NBI vs. traditional colposcopy with the green-filter should be further scheduled.

In conclusion, ME-NBI did not show a higher detection ability than colposcopy for patients with CIN2+. As ME-NBI was associated with excellent patient acceptability, further verification in a large-scale controlled trial is warranted.

## Figures and Tables

**Figure 1 jcm-10-04753-f001:**
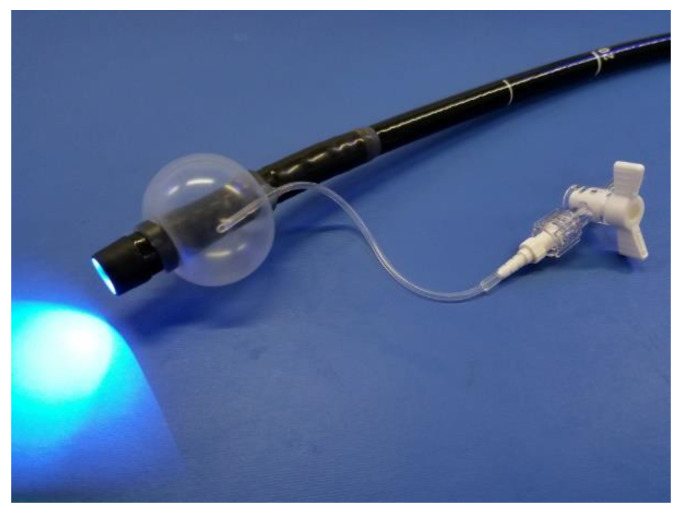
Flexible magnifying endoscopy with narrow band imaging (ME-NBI) with a soft black hood and an occlusion balloon.

**Figure 2 jcm-10-04753-f002:**
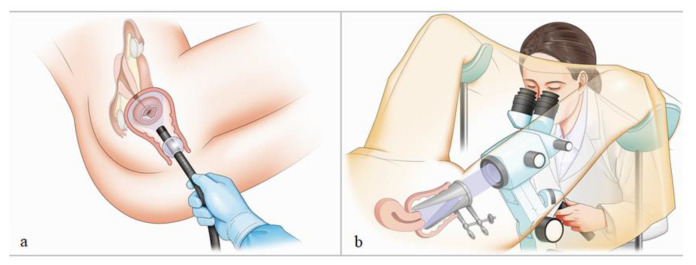
Schematic illustration of patient positioning: ME-NBI (**a**) and colposcopy (**b**).

**Figure 3 jcm-10-04753-f003:**
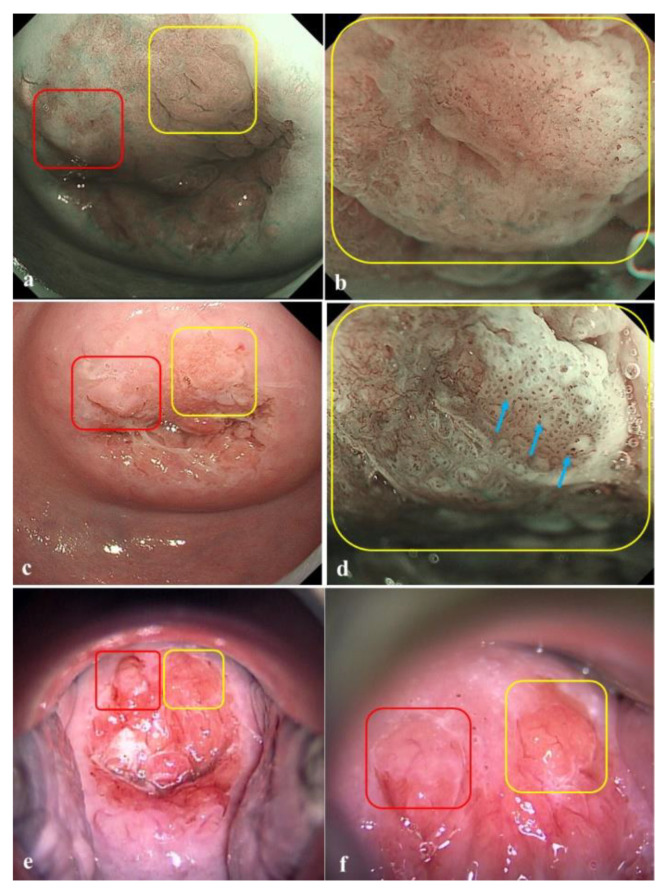
A representative case of cervical intraepithelial neoplasia grade 3 (CIN3) detected by ME-NBI but not with colposcopy. In narrow band imaging (NBI), an area with thin-white epithelium at the 1–2 o’clock position was observed in the cervix (yellow box) (**a**). ME-NBI revealed atypical vessels in the area with thin-white epithelium (**b**). Acetic acid endoscopy in white light imaging did not show acetowhite epithelium in the same area (**c**). ME-NBI with acetic acid contrasted the area with thin-white epithelium and atypical vessels, which appeared with dilatation and a meandering, irregular arrangement, and caliber change (blue arrows). This lesion was confirmed histologically as CIN3 (**d**). Colposcopy with acetic acid application showing a thin acetowhite lesion (Grade 1) at the 11 o’clock position. Biopsies confirmed a histological diagnosis of CIN1 (red box) (**e**). Magnified images (**f**) of those shown in (**e**).

**Figure 4 jcm-10-04753-f004:**
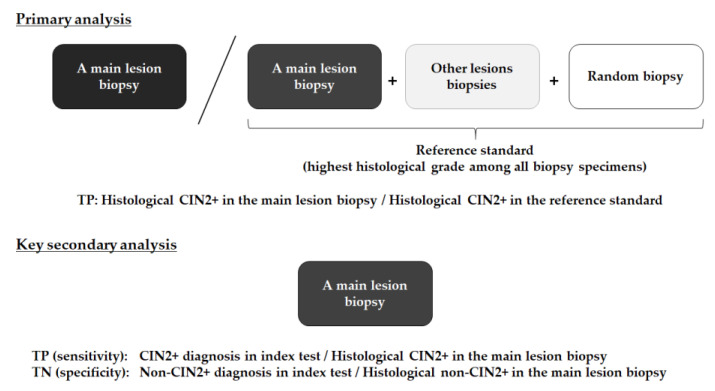
Definition of primary and key secondary analyses. TP, true positive; TN, true negative; CIN2+, cervical intraepithelial neoplasia grade 2 or worse.

**Figure 5 jcm-10-04753-f005:**
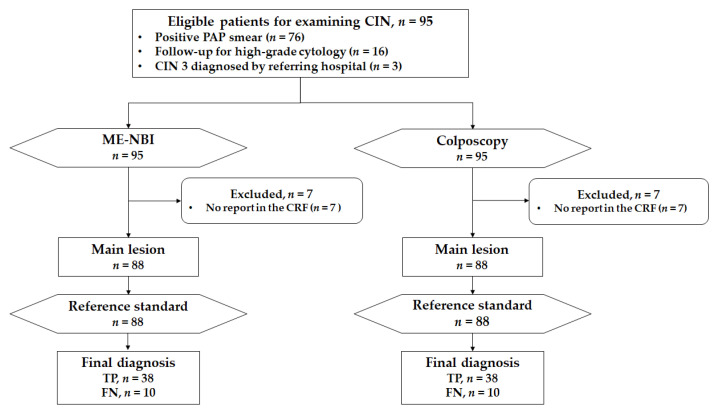
A STARD flow chart for primary analysis: detection ability for patients with CIN2+. TP, true positive; FN, false negative.

**Figure 6 jcm-10-04753-f006:**
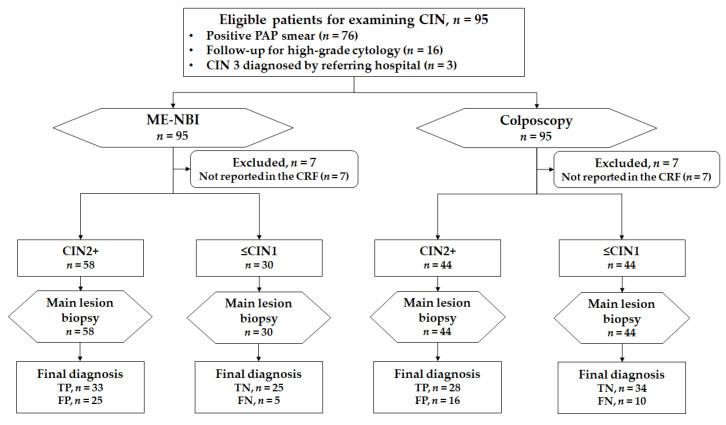
A STARD flow chart for a key secondary analysis: diagnostic ability for histological CIN2+ lesions. TP, true positive; TN, true negative; FP, false positive; FN, false negative.

**Table 1 jcm-10-04753-t001:** Patients’ baseline demographics and clinical characteristics.

Total Number of Patients	88
Median age, years (range)	40 (21–67)
Indications, *n* (%)	
Positive Pap smear	72
High-grade cytology on follow-up	13
Definitive CIN3	3
HPV infection, *n* (%)	
Positive	25
Negative	5
Not examined	58
HPV vaccination, *n* (%)	
Received	5
Not received	83
Institution, *n* (%)	
Kochi Red Cross Hospital	60
Kagawa University Hospital	25
OICI	3
Final histological diagnosis in the reference standard, *n* (%)	
Non-cancerous lesion	8
CIN1	32
CIN2	13
CIN3	34
Microinvasive carcinoma	1

CIN1, 2, and 3, cervical intraepithelial neoplasia grade 1, 2, and 3; HPV, human papillomavirus; OICI, Osaka International Cancer Institute.

**Table 2 jcm-10-04753-t002:** Comparison of the primary and key secondary endpoints between ME-NBI and colposcopy.

Measures	ME-NBI (*n* = 88)	Colposcopy (*n* = 88)	*p* Value
Detection rate for patients with CIN2+	79.2 (65.7–88.3)	79.2 (65.7–88.3)	1.000
Diagnostic ability for histological CIN2+ lesions			
Sensitivity	86.8 (72.7–94.3)	73.7 (58.0–85.0)	0.15
Specificity	50.0 (36.6–63.4)	68.0 (54.2–79.2)	0.067
Overall accuracy	65.9 (55.5–75.0)	70.5 (60.2–79.0)	0.517
Proportion of patients with visible whole circumferential transitional zones, %	97.7	90.9	0.031
Proportion of patients with visible external uterine orifice, %	96.6	89.8	0.073
Incidence of Grade 2 adverse events	0	0	
Acceptability of patients, grade, *n* (%)			
Discomfort (*n* = 87)			
1	61 (69.3)	9 (10.2)	
2	23 (26.1)	14 (15.9)	
3	2 (2.3)	14 (15.9)	
4	1 (1.1)	24 (27.3)	
5	0	26 (29.5)	
Embarrassment (*n* = 88)			
1	45 (51.1)	20 (22.7)	
2	29 (33.0)	11 (12.5)	
3	8 (9.1)	17 (19.3)	
4	5 (5.7)	16 (18.2)	
5	1 (1.1)	24 (27.2)	
Preference to next examination (*n* = 86), *n* (%)	65 (75.6)	2 (2.3)	<0.001

ME-NBI, magnifying endoscopy with narrow band imaging; CIN2+, cervical intraepithelial neoplasia grade 2 or worse.

## Data Availability

Data are not publicly available due to protection of personal data and medical confidentiality.

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
