# Peer review of "Flexible Magnifying Endoscopy with Narrow Band Imaging for Diagnosing Uterine Cervical Neoplasms: A Multicenter Prospective Study"

_jcm, 2021, doi:10.3390/jcm10204753_

Round 1

Reviewer 1 Report

I read with interest the manuscript titled “Flexible Magnifying Endoscopy with Narrow Band Imaging for Diagnosing Uterine Cervical Neoplasms: A Multicenter prospective Study.” The authors conducted a prospective study for diagnosing uterine cervical neoplasm with NBI. The usability of NBI for neoplasms other than the GI tract was unknown. This study showed an adequate diagnostic performance and higher acceptability of endoscopy for diagnosing uterine cervical neoplasms. This was also a meaningful study to establish a process for applying technology to another research field. The authors should be congratulated on this challenging procedure. The manuscript is well written. However, there is a big problem in terms of the Clinical Trials Act. The following may be taken into consideration.

Comments

#1 3.2. Outcome

The authors performed the comparative analysis between the detection rates in NBI and colposcopy, and there was no significant difference. Even though the values were the same, the authors couldn't mention an equivalence. The result form analysis just showed the detection rate of NBI was “not” significantly higher than that of colposcopy. The authors should edit the relevant sentences at abstract, results and discussion.

#2.

The authors use endoscopy for a condition that was not approved in this trial; off-label use. In April 2018, Clinical Trials Act was enacted in Japan (Japanese Journal of Clinical Oncology, 2020, 50(4)399–404). I believe this trial should have been conducted under the Clinical Trials Act. How to clear this issue?

If a third party points out this problem after publication, the paper can be withdrawn. 

#3

Please add “highest histological grade among all biopsy specimens” in Figure 4 to make it understand easier.

Author Response

Response to REVIEWERS' COMMENTS:

Reviewer 1.
I read with interest the manuscript titled “Flexible Magnifying Endoscopy with Narrow Band Imaging for Diagnosing Uterine Cervical Neoplasms: A Multicenter prospective Study.” The authors conducted a prospective study for diagnosing uterine cervical neoplasm with NBI. The usability of NBI for neoplasms other than the GI tract was unknown. This study showed an adequate diagnostic performance and higher acceptability of endoscopy for diagnosing uterine cervical neoplasms. This was also a meaningful study to establish a process for applying technology to another research field. The authors should be congratulated on this challenging procedure. The manuscript is well written. However, there is a big problem in terms of the Clinical Trials Act. The following may be taken into consideration.

Comments

#1 3.2. Outcome

The authors performed the comparative analysis between the detection rates in NBI and colposcopy, and there was no significant difference. Even though the values were the same, the authors couldn't mention an equivalence. The result form analysis just showed the detection rate of NBI was “not” significantly higher than that of colposcopy. The authors should edit the relevant sentences at abstract, results and discussion.

Response

Thank you for your precise comments.

Abstract: We changed the sentence ‘ME-NBI showed promising detection ability for patients with CIN2+ and better patient acceptability.’ to ‘ME-NBI did not show detection ability over colposcopy for patients with CIN2+, whereas showed better patient acceptability.’

Results: We changed the sentence ‘The detection rate of ME-NBI for patients with CIN2+ was 79.2% (95% CI: 65.7%–88.3%), which was the same as with colposcopy (79.2%, P = 1.000)’ to ‘The detection rate of ME-NBI for patients with CIN2+ was 79.2% (95% CI: 65.7%–88.3%), which was not significantly higher than that of colposcopy (79.2%, P = 1.000).’

Discussion: We revised two following sentences:

‘First, ME-NBI showed comparable diagnostic ability with that of colposcopy for patients with CIN2+.’

→‘First, ME-NBI did not significantly show higher detection ability than colposcopy for patients with CIN2+.’

‘In conclusion, ME-NBI showed comparable detection ability for patients with CIN2+ with that of colposcopy.’

→‘In conclusion, ME-NBI did not show detection ability over colposcopy for patients with CIN2+.’

#2.

The authors use endoscopy for a condition that was not approved in this trial; off-label use. In April 2018, Clinical Trials Act was enacted in Japan (Japanese Journal of Clinical Oncology, 2020, 50(4)399–404). I believe this trial should have been conducted under the Clinical Trials Act. How to clear this issue?

If a third party points out this problem after publication, the paper can be withdrawn. 

Response

Thank you for your valuable and appropriate comments.

Allowance for use of gastroscopy in the uterine cervix was confirmed in the ministry of health, labour, and welfare office, and it was deliberated and approved by the institutional review board.

As readers can understand, we added the following sentence in the end of ‘2.1. Study Design’; Use of gastroscope for diagnosis of the uterine cervix neoplasm was approved by the institutional review board.

#3

Please add “highest histological grade among all biopsy specimens” in Figure 4 to make it understand easier.

Response

As you suggest, we added “highest histological grade among all biopsy specimens” in Figure 4.

Reviewer 2 Report

The manuscript is well designed, the methods adeguate and the results well descripted. The limits of the tecnique should be discussed: impossibility to evaluate the vaginal fornix; moreover, as regards the description of vascularization, a comparison between ME-NBI vs traditional colposcopy with the green-filter is lacking.

Author Response

Response to REVIEWERS' COMMENTS:

Reviewer 2.

The manuscript is well designed, the methods adeguate and the results well descripted. The limits of the tecnique should be discussed: impossibility to evaluate the vaginal fornix; moreover, as regards the description of vascularization, a comparison between ME-NBI vs traditional colposcopy with the green-filter is lacking.

Response

Thank you for your valuable comments.

Q1. ‘impossibility to evaluate the vaginal fornix’:

As described in the Discussion, ME-NBI, with our developed vagina-occluding balloon, enabled examiners to secure a visual field that included the entire uterine cervix, without special skills.

We revised the sentence as follows: ‘ME-NBI, with our developed vagina-occluding balloon, enabled examiners to secure a visual field that included the entire uterine cervix and vaginal fornix by air insufflation via endoscope, without special skills.’

Q2. ‘comparison between ME-NBI vs traditional colposcopy with the green-filter’

As you suggest, we added the following sentence in the limitation; Third, the colposcopic green-filter for evaluating neoplastic vascularization was not introduced in this study because it was not clinically used in our institutions. A comparative study between ME-NBI vs. traditional colposcopy with the green-filter should be further scheduled.’
